# The Optimum Monitoring Location of Pressure in Water Distribution System

**Sanghyun Jun [1] and Hyuk Jae Kwon [2,*]**

[1]  Infra division, POSCO E&C, 241 Incheon tower daero, Incheon 22009, Korea; clays@poscoenc.com
[2]  Department of Civil Engineering, Cheongju University, Chungbuk 28503, Korea
*  Correspondence: hjkwon@cju.ac.kr; Tel.: +82-43-229-8473

**Abstract:** This study proposes two methods for the determination of optimum monitoring locations of pressure changes in a water distribution system. A sensitivity analysis method is used to calculate the pressure change in a junction due to the change in demand at other junctions. A pressure contribution analysis method is used to calculate the summation of pressure contribution of a junction due to the change in demand at another junction. These methods are applied to a small sample pipe network, pilot plant, and small distribution system for verification. Furthermore, unsteady analysis of the sample pipe network and an experiment in the pilot plant are conducted to verify the availability and accuracy of the proposed methods. To verify the methods, leakage at J-55 was artificially produced at the pilot plant. The pressure change was measured at five different combination groups of sensor locations. From the results, it was found that the top ranked group of sensor locations, J-116, J-140, J-22, and J-68, had the highest pressure contributions and sensitivity. The results of the newly developed methods for the determination of monitoring locations are in good agreement with the results of the unsteady analysis. Finally, the proposed methods are applied to a real distribution system of a small city as a test bed. It is found that the proposed methods for determining the monitoring locations of pressure changes in the water distribution system are useful and effective.

**Keywords:** monitoring location; water distribution system; pressure change; pressure contribution; pressure sensitivity

## 1. Introduction

Real-time pressure monitoring is important to detect leakage and abnormal pressure changes for the efficient management of water distribution systems. Water pressure in a distribution system should be cautiously monitored since it is a very important matter for maintenance. An optimal determination method for monitoring locations is necessary because it is impossible to install pressure gauges at every place in a water distribution system. From the results of further development, it will be possible to take good care of junctions that are more sensitive and more contributive than other junctions, as the determination method can detect the abnormal pressure changes and leakage.

Pudar and Liggett [1] presented basic research and developments about leakage sensing. They  proposed that the leakage detection problem can be solved by the back-tracing method using pressure gauges or flow meters. Leakage can be detected by a comparison between the pressure measurement according to the changes in flow rate in the orifice and pressure changes due to real leakage measurement. A sensitivity matrix can be used for the determination of monitoring locations. The sensitivity matrix showed that leakage amount and location could be sensitively related to data size and accuracy of pressure measurement.

Liggett and Chen [2] proposed a monitoring method using unsteady reverse analysis in the water distribution system. They used the method of characteristics for unsteady analysis and also used time-lagged calculations.

Misiunas et al. [3] proposed a method for breakage or leakage detection using unsteady analysis. They showed that the leakage location can be detected by back-tracing the velocity of an unsteady pressure wave. The size of the leakage could be estimated by the unsteady pressure wave and the particular shape of pressure wave due to a sudden break.

Vitkovsky et al. [4] suggested the inverse transient analysis method to estimate the leakage location and amount by the unsteady pressure wave. The shape of the unsteady pressure wave is important for the successive application of inverse transient analysis. They conducted leakage experiments in the laboratory to verify the method. The experimental results showed that the leakage amount was successfully estimated and the leakage location was correctly determined.

Jun et al. [5] developed an algorithm to determine the monitoring location based on the effective index matrix of a water distribution system. The installation of pressure gauges can be restricted as initial investigation can be exceeded. The proposed method estimated the location of online pressure gauges by conducting a column search on the effective index matrix. Moreover, the method was applied to the Cherry Hill distribution system to verify its efficiency.

Cheong et al. [6] suggested a method to determine the optimum location of pressure gauges using entropy theory. In their study, entropy theory was applied to overcome the shortcomings of previous methods. For example, previous methods were difficult to apply for a specific area, which does not have systematic management, as these methods need to verify and calibrate the measured data. Furthermore, most previous works focused on the determination of junctions that can minimize the measurement cost.

In this study, two optimum determination methods of pressure gauge locations are proposed, namely the sensitivity analysis and pressure contribution analysis. Experiments were conducted to verify the methods by measuring the pressure changes due to leakage in a real-size pilot plant. The relatively more sensitive monitoring locations according to pressure changes due to leakage were selected and compared with the selected junctions from the results of the sensitivity analysis and pressure contribution analysis. The sensitivity analysis and pressure contribution analyses can be used as determination methods for monitoring locations to detect the leakage or abnormal situation in a water distribution system. These methods can be used to determine whether or where the specific facility should be installed in the water distribution system.

## 2. Determination of Monitoring Location

Figure 1 shows a sample pipe network with 15 pipes, 11 junctions, and 1 distributing reservoir. Tables 1 and 2 show the properties of the pipes and the demand at junctions. L is length of pipe, J is junction, D is pipe diameter, C is Hazen-Williams coefficient. Q is demand at the junction.

**Table 1.** Pipe properties of sample pipe network.

| Pipe | 1 | 2 | 3 | 4 | 5 | 6 | 7 | 8 | 9 | 10 | 11 | 12 | 13 | 14 | 15 |
|------|-----|-----|-----|-----|-----|-----|-----|-----|-----|-----|-----|-----|-----|-----|-----|
| L (m) | 400 | 400 | 300 | 350 | 350 | 300 | 300 | 400 | 450 | 300 | 250 | 200 | 200 | 200 | 150 |
| D (cm) | 30 | 35 | 35 | 30 | 30 | 35 | 30 | 35 | 30 | 30 | 30 | 25 | 25 | 25 | 25 |
| C | 100 | 100 | 100 | 100 | 100 | 100 | 100 | 100 | 100 | 100 | 100 | 100 | 100 | 100 | 100 |

**Table 2.** Demand at junctions.

| Junction | 1 | 2 | 3 | 4 | 5 | 6 | 7 | 8 | 9 | 10 | 11 | 12 |
|----------|-----|-----|------|-----|------|------|------|------|------|-----|-----|------|
| Q (m$^3$/s) | 0.0 | 0.0 | 0.05 | 0.0 | 0.07 | 0.07 | 0.05 | 0.06 | 0.05 | 0.1 | 0.1 | 0.09 |

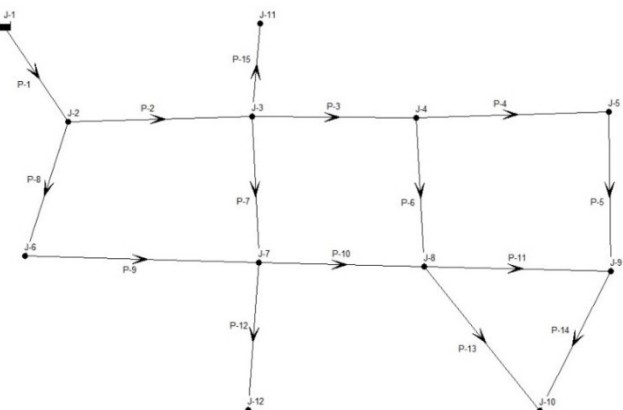

**Figure 1.** Sample pipe network.

### 2.1. Sensitivity Analysis According to Change in the Hazen–Williams Coefficient

Firstly, the pressure head at junctions and the flow rate at pipes are estimated without any change in demand. Sensitivity analysis according to the change in the Hazen–Williams coefficient is conducted to determine the monitoring location for leakage detection. For example, steady analysis is conducted by changing the Hazen–Williams coefficient of pipe 1 from 100 to 120. At the same time, the Hazen–Williams coefficients of other pipes are maintained as 100—the original coefficient. Sensitivity analysis is conducted using the following equation:

$$S_j = \sum_{j=1}^{k} \frac{\Delta h_j}{h_j} / k \tag{1}$$

where $k$ is the number of pipes, $j$ is the junction number, $h_j$ is the original pressure, and $\Delta h_j$ is the pressure change at each junction due to the change in the Hazen–Williams coefficient. Therefore, the pressure change due to the change in the Hazen–Williams coefficient is summed up and averaged by the number of pipes. The rank of priority for the junctions is estimated by the pressure sensitivity. Table 3 shows the results of the sensitivity analysis.

**Table 3.** Results of sensitivity analysis according to the change in the Hazen–Williams coefficient.

| Rank | 1 | 2 | 3 | 4 | 5 | 6 | 7 | 8 | 9 | 10 | 11 |
|------|---|---|---|---|---|---|---|---|---|----|----|
| Junction | 5 | 10 | 9 | 8 | 4 | 12 | 11 | 7 | 3 | 6 | 2 |

### 2.2. Pressure Contribution Analysis According to the Change in Demand

The pressure change at a junction due to the change in demand can cause a pressure change at another junction. Therefore, the pressure contribution at each junction can be quantitatively measured. In this study, pressure contribution is estimated by the averaged pressure change. The steady-state network analysis is conducted by creating an additional demand at a junction. At this point, the original pressure at each junction will change. The pressure difference between $i$th junction and $j$th junction divided by the original pressure at the $i$th junction is averaged and compared with each other. As shown in Equation (2), the pressure contribution of one junction to another can be quantitatively estimated and compared with each other.

$$C_i = \left[ \sum_{j=1}^{k} \left| \frac{(h_i - h_j)}{h_i} \right| \right] / k \tag{2}$$

where $k$ is the number of pipes, $i,j$ is the junction number, $h_i$ is the original pressure head at the $i$ junction and $h_j$ is the pressure head at the $j$ junction due to the change in demand at the $i$ junction.

This equation is used to estimate how the pressure change at a junction due to demand can affect the pressure at another junction. The steady-state analysis is conducted by changing the demand at each junction. At this time, analysis results such as the pressure at each junction should be changed as the demand at the specific junction is changed. Therefore, the pressure change at the $j$ junction according to the sudden change in demand at the $i$ junction to the original pressure at the $i$ junction can be summed up and averaged by the number of junctions; this is called the pressure contribution of the $i$ junction. The pressure change can be estimated at each time when the demand at a junction is changed. Therefore, the number of total junctions should be the size of the pressure contribution matrix. The amount of change due to demand is 0.02 m$^3$/s in the sample pipe network. Figure 2 shows the flowchart for the determination of monitoring locations using pressure contribution analysis. Table 4 shows the results of the estimation. The results show that the pressure contribution of J-10 was the highest in the sample pipe network.

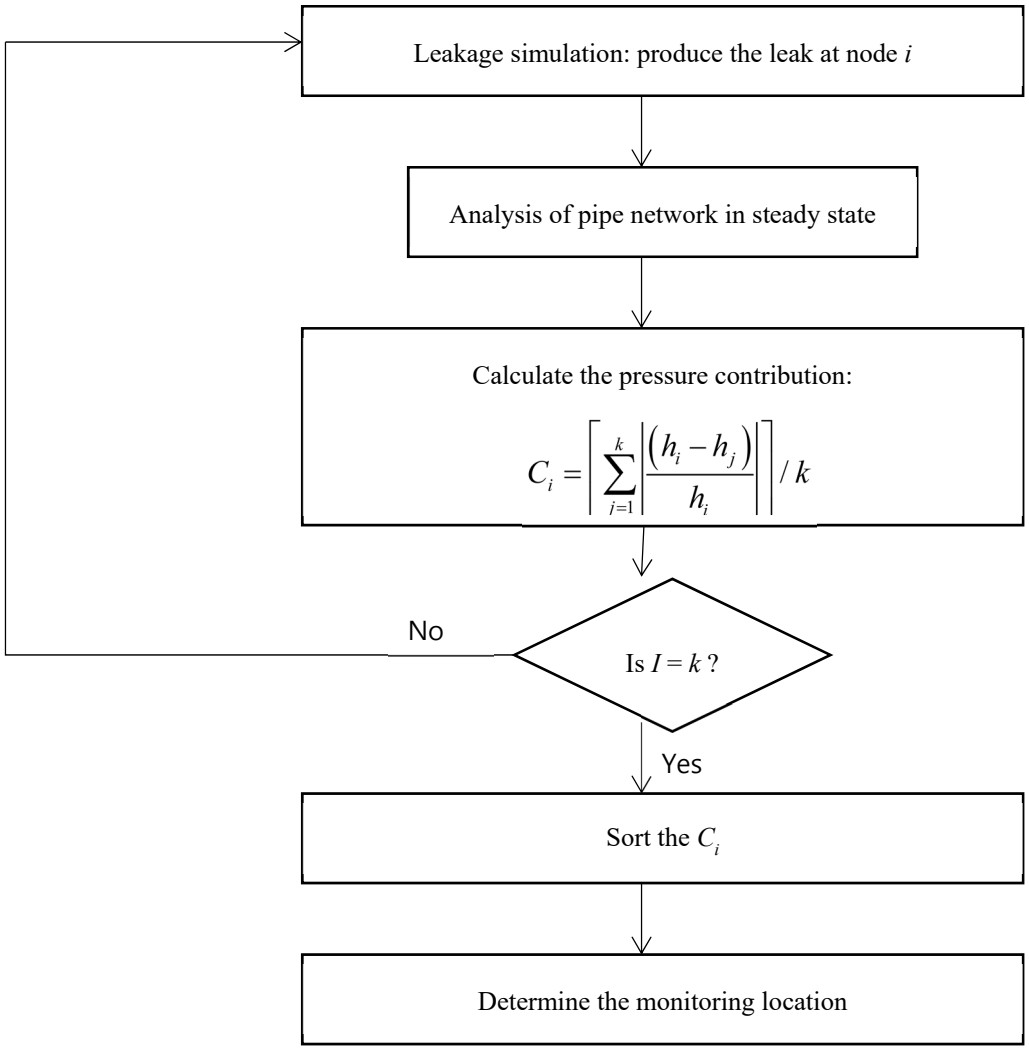

**Figure 2.** Flowchart for the determination of monitoring location using pressure contribution analysis.

**Table 4.** Results of pressure contribution analysis.

| Rank | 1 | 2 | 3 | 4 | 5 | 6 | 7 | 8 | 9 | 10 | 11 |
|---|---|---|---|---|---|---|---|---|---|---|---|
| Junction | 10 | 9 | 5 | 8 | 4 | 12 | 7 | 11 | 3 | 6 | 2 |

*2.3. Pressure Sensitivity Analysis According to the Change in Demand*

Pressure sensitivity analysis estimates the pressure change at a junction according to the change in demand. The ratio of pressure change to the original pressure at each junction is summed up and averaged by the number of junctions as the pressure sensitivity of a junction. Therefore, the number of junctions should be the size of the sensitivity matrix as the steady-state analysis is conducted whenever the demand conditions at each junction are changed. Pressure sensitivity, represented by Equation (3), is the averaged pressure differences of the *i*th junction due to the change in demand. As shown in Equation (3), the pressure sensitivity of each junction can be quantitatively estimated and compared with each other.

$$S_i = \left[ \sum_{i=1}^{k} \left| \frac{(h_i - h'_i)}{h_i} \right| \right] / k \tag{3}$$

where $k$ is the number of calculations, $i$ is the junction number, $h_i$ is the original pressure at $i$ junction and $h'_i$ is the changed pressure at $i$ junction according to the change in demand at a specific junction. Table 5 shows the results of the sensitivity analysis according to the change in demand.

**Table 5.** Results of sensitivity analysis according to the change in demand.

| Rank | 1 | 2 | 3 | 4 | 5 | 6 | 7 | 8 | 9 | 10 | 11 |
|------|---|---|---|---|---|---|---|---|---|----|----|
| Junction | 9 | 5 | 10 | 8 | 4 | 12 | 7 | 11 | 3 | 6 | 2 |

Table 6 and Equations (4) and (5) can well explain the sensitivity analysis and pressure contribution analysis. The pressure contribution of J-1 can be calculated by Equation (4) and the pressure sensitivity of J-1 is calculated by Equation (5). The number of junctions is three and the number of calculations is also three.

$$C_1 = \left[ \frac{(P_1 - A) + (P_1 - D) + (P_1 - G)}{P_1} \right] / 3 \tag{4}$$

$$S_1 = \left[ \frac{(P_1 - A) + (P_1 - B) + (P_1 - C)}{P_1} \right] / 3 \tag{5}$$

**Table 6.** Examples of pressure at each junction according to simulation number.

| Simulation No. | Pressure at J-1 | Pressure at J-2 | Pressure at J-3 |
|:---:|:---:|:---:|:---:|
| 0 | $P_1$ | $P_2$ | $P_3$ |
| 1 | A | D | G |
| 2 | B | E | H |
| 3 | C | F | I |

## 3. Determination of Monitoring Location Using Unsteady Analysis

*3.1. Unsteady Analysis of a Water Distribution System*

Unsteady analysis of a pipe network shows the pressure and flow rate at any location as a function of time. Both the continuity equation (Equation (6)) and the equation of motion (Equation (7)) should be used in unsteady analysis. Although various methods were introduced to solve these two equations, the results of all methods were similar. The method of characteristics [7–9] was selected for the present study as it is numerically stable, accurate, convenient to use, and has a short computation time. The equation of motion and the continuity equation for the method of characteristics can be summarized as follows [9,10]:

$$\frac{\partial Q}{\partial t} + gA \frac{\partial H}{\partial x} + \frac{f}{2DA} Q|Q| = 0 \tag{6}$$

$$\frac{c^2}{gA}\frac{\partial Q}{\partial x} + \frac{\partial H}{\partial t} = 0 \tag{7}$$

where $Q$ represents the flow rate, $H$ is the pressure head, $A$ is the cross-sectional area of the pipe, $c$ is the speed of pressure wave, and $f$ is the Darcy–Weisbach friction coefficient. For the present study, only friction loss is considered for head loss. Therefore, it is assumed that minor losses are negligible for the present computations.

Next, let us consider $L_1$ as the equation of motion and $L_2$ as the continuity equation. A linear combination of equations, $L = L_1 + \lambda L_2$ (where $\lambda$ is the Lagrangian multiplier) yields Equation (8).

$$\left(\frac{\partial Q}{\partial t} + \lambda c^2 \frac{\partial Q}{\partial x}\right) + \lambda g A \left(\frac{\partial H}{\partial t} + \frac{1}{\lambda}\frac{\partial H}{\partial x}\right) + \frac{f}{2DA}Q|Q| = 0 \tag{8}$$

It is noted that $H = H(x,t)$ and $Q = Q(x,t)$. Thus, the total derivatives may be written by the chain rule as:

$$\frac{dQ}{dt} = \frac{\partial Q}{\partial t} + \frac{\partial Q}{\partial x}\frac{dx}{dt} \tag{9}$$

$$\frac{dH}{dt} = \frac{\partial H}{\partial t} + \frac{\partial H}{\partial x}\frac{dx}{dt} \tag{10}$$

$\lambda$ is defined by comparing Equation (8) with Equations (9) and (10). In Equation (9), if $\frac{dx}{dt}$ is equal to $\lambda c^2$, then the first parenthesis of Equation (8) is exactly the same as Equation (9). In Equation (10), if $\frac{dx}{dt}$ is equal to $\frac{1}{\lambda}$, then the second parenthesis of Equation (8) is exactly the same as Equation (10). Therefore, $\frac{1}{\lambda} = \frac{dx}{dt} = \lambda c^2$.

$$\lambda = \pm\frac{1}{c} \tag{11}$$

By using these equations, Equation (8) can be rewritten as:

$$\frac{dQ}{dt} + \frac{gA}{c}\frac{dH}{dt} + \frac{f}{2DA}Q|Q| = 0 \tag{12}$$

$$\frac{dQ}{dt} - \frac{gA}{c}\frac{dH}{dt} + \frac{f}{2DA}Q|Q| = 0 \tag{13}$$

Equation (12) is valid if $\frac{dx}{dt} = c$. Furthermore, Equation (13) is valid if $\frac{dx}{dt} = -c$.

The finite difference equations of Equations (12) and (13) can be written as follows:

$$(Q_i^{n+1} - Q_{i-1}^n) + \frac{gA}{c}(H_i^{n+1} - H_{i-1}^n) + \frac{f\Delta t}{2DA}Q_{i-1}^n|Q_{i-1}^n| = 0 \tag{14}$$

$$(Q_i^{n+1} - Q_{i+1}^n) - \frac{gA}{c}(H_i^{n+1} - H_{i+1}^n) + \frac{f\Delta t}{2DA}Q_{i+1}^n|Q_{i+1}^n| = 0 \tag{15}$$

where the superscript $n + 1$ represents the unknowns. The well-known stability and convergence condition must be satisfied as:

$$\frac{\Delta t}{\Delta x} \leq \frac{1}{c} \tag{16}$$

In the present study, $c\Delta t/\Delta x = 1$ is used for the stability condition for the entire computation.

### 3.2. Results of Unsteady Analysis

Unsteady analysis was performed by the method of characteristics with the assumption of a sudden change in demand at each junction. It was assumed that an additional demand of 0.02 m$^3$/s occurred at each junction. At the first calculation, it was assumed that a demand of 0.02 m$^3$/s suddenly occurred at J-1. Even though J-1 does not have demand, unsteady analysis was performed assuming that a demand abruptly occurred at J-1. At the second calculation, unsteady analysis was performed assuming that an additional demand of 0.02 m$^3$/s occurred at J-2, which has a usual demand of

0.08 m$^3$/s. The sensor location can be determined by sensitivity analysis for the real leakage because a sudden additional demand can be considered as leakage at a specific junction. Table 7 shows the pressure change at each junction after unsteady analysis. A total of 11 unsteady analyses were performed as shown in Table 7, as J-14 is considered as a reservoir and neglected for the calculation. Figure 3 shows the pressure oscillations at (a) J-7 and (b) J-10. The analysis conditions $\Delta t = 0.008$ s, $\Delta x = 10$ m, $c = 1250$ m/s (Courant Number = 1.0), and Darcy–Weisbach coefficient = 0.015 were applied. As shown in the figures, the pressure at J-10 responds more sensitively for the change in demand. A pressure change is defined as the difference between the maximum pressure and the minimum pressure due to a sudden change in demand. In Table 7, the first row shows each junction and the next rows show the pressure change due to a sudden change in demand at each junction. The first column shows the original piezometric head obtained from the steady-state analysis.

**Table 7.** Pressure change at each junction due to a sudden change in demand.

| Piezometric Head (m) | No. | J-2 (m) | J-3 (m) | J-4 (m) | J-5 (m) | J-6 (m) | J-7 (m) | J-8 (m) | J-9 (m) | J-10 (m) | J-11 (m) | J-12 (m) |
|---|---|---|---|---|---|---|---|---|---|---|---|---|
| 230.27 | 2 | 14.79 | 14.74 | 21.43 | 24.66 | 12.13 | 12.15 | 16.07 | 22.58 | 21.03 | 18.02 | 21.29 |
| 213.53 | 3 | 11.9 | 22.29 | 25.04 | 23.49 | 18.50 | 17.44 | 20.79 | 29.87 | 25.22 | 26.63 | 20.06 |
| 210.04 | 4 | 13.19 | 19.58 | 24.74 | 31.49 | 19.32 | 21.31 | 24.60 | 31.10 | 30.70 | 24.19 | 25.53 |
| 209.23 | 5 | 15.99 | 21.56 | 27.33 | 47.9 | 27.85 | 24.86 | 31.48 | 31.69 | 33.11 | 26.98 | 28.70 |
| 222.60 | 6 | 19.15 | 21.68 | 22.35 | 26.61 | 29.01 | 20.71 | 19.82 | 24.39 | 21.01 | 27.11 | 23.88 |
| 212.66 | 7 | 15.14 | 20.07 | 22.79 | 30.58 | 21.41 | 23.64 | 22.09 | 26.88 | 25.14 | 26.08 | 26.00 |
| 209.53 | 8 | 13.55 | 21.95 | 25.32 | 28.15 | 21.09 | 18.95 | 24.78 | 32.73 | 23.91 | 20.08 |
| 209.23 | 9 | 16.80 | 26.79 | 31.85 | 32.96 | 30.26 | 18.91 | 28.00 | 37.48 | 37.18 | 32.29 | 28.63 |
| 209.23 | 10 | 15.06 | 22.12 | 23.32 | 26.80 | 23.45 | 20.26 | 21.28 | 27.51 | 25.92 | 23.72 | 22.34 |
| 213.35 | 11 | 15.37 | 26.64 | 25.20 | 22.72 | 18.08 | 20.99 | 23.64 | 30.53 | 27.82 | 33.31 | 18.93 |
| 212.47 | 12 | 17.20 | 19.77 | 27.55 | 34.98 | 21.07 | 26.67 | 27.43 | 29.55 | 25.86 | 24.07 | 29.93 |

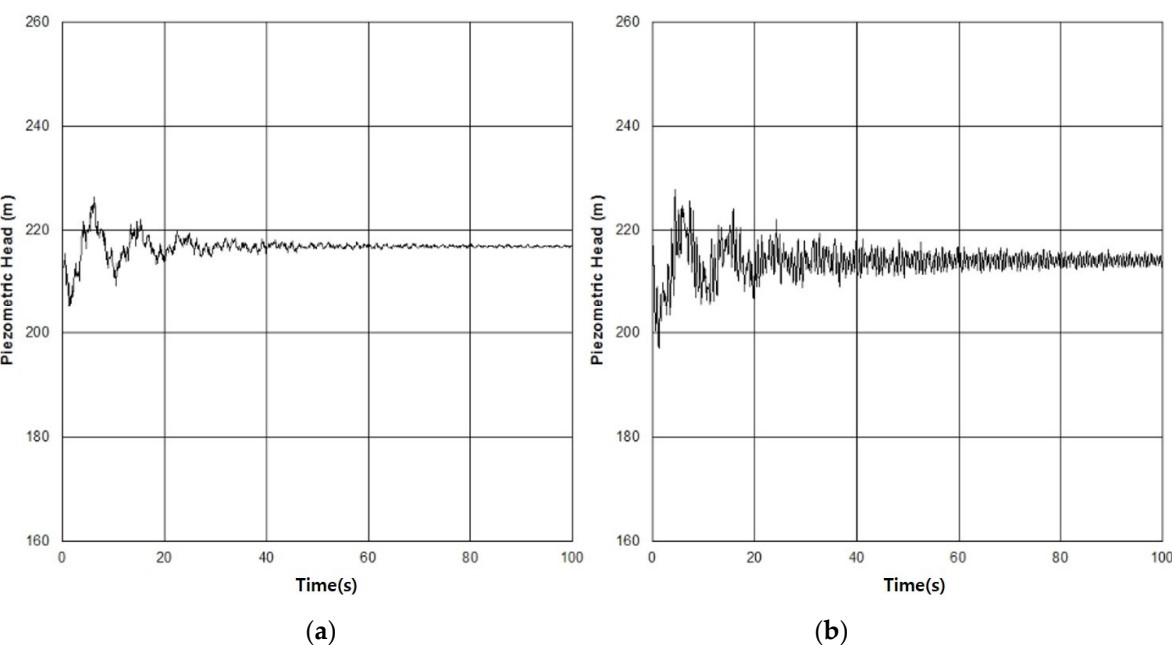

**Figure 3.** Pressure oscillations at (**a**) J-7 and (**b**) J-10.

### 3.3. Determination of Monitoring Location Using the Results of Unsteady Analysis

Unsteady flow was produced by changing the boundary condition by introducing a sudden change in demand. A sudden change in demand can cause a pressure change that can be observed at any place in the piping system. The results of unsteady analysis can be used for the determination of monitoring locations. Firstly, pressure height due to a sudden change in demand should be estimated. The ratio of pressure height to the original pressure at each junction is summed up and averaged by the number of junctions to compute pressure sensitivity. In this study, sensitivity was prescribed as shown in Equation (2) and estimated. Table 8 shows the results of the estimation suggesting the top

rank of sensor location. The first rank of sensor location is determined as J-5, the second rank is J-9, and the third rank is J-10. Table 9 shows the results of pressure contribution analysis using the results of unsteady analysis. As shown in Table 9, the first rank of sensor location is determined as J-9, the second rank is J-5, and the third rank is J-12.

**Table 8.** Results of sensitivity analysis using the results of unsteady analysis.

| Rank | 1 | 2 | 3 | 4 | 5 | 6 | 7 | 8 | 9 | 10 | 11 |
|---|---|---|---|---|---|---|---|---|---|---|---|
| Junction | 5 | 9 | 10 | 11 | 4 | 12 | 8 | 3 | 6 | 7 | 2 |

**Table 9.** Results of pressure contribution analysis using the results of unsteady analysis.

| Rank | 1 | 2 | 3 | 4 | 5 | 6 | 7 | 8 | 9 | 10 | 11 |
|---|---|---|---|---|---|---|---|---|---|---|---|
| Junction | 9 | 5 | 12 | 4 | 8 | 11 | 7 | 10 | 6 | 3 | 2 |

## 4. Applications to the Pilot Plant and Test Bed

### 4.1. Pilot Plant

Figure 4 shows the front view of the pilot plant. The pilot plant is constructed with one pump, 144 junctions, and 179 cast iron pipes, which have the same diameter of 0.1 m. The length of the long side is approximately 60 m.

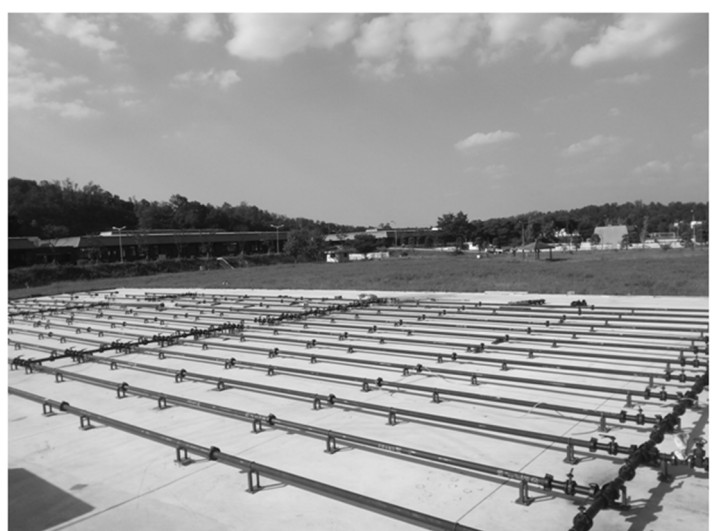

**Figure 4.** Front view of pilot plant.

The proposed methods for the determination of monitoring locations were confirmed by unsteady analysis. It was proved that the methods are outstanding because of their short calculation time and simplicity. In this study, the monitoring location in the pilot plant was determined by pressure contribution analysis and sensitivity analysis as shown in Table 10. The first, second, and third ranked monitoring locations are J-143, J-142, and J-141, respectively. However, J-143, J-142, and J-141 are the very last down streams of the plant. Therefore, J-116 should be determined as the top ranked monitoring location in this pilot plant.

**Table 10.** Rank of monitoring location for pilot plant.

| Method | 1st Rank | 2nd Rank | 3rd Rank | 4th Rank | 5th Rank | 6th Rank | 7th Rank | 8th Rank | 9th Rank |
|---|---|---|---|---|---|---|---|---|---|
| Pressure contribution | 143 | 142 | 141 | 116 | 140 | 22 | 139 | 68 | 65 |
| Pressure sensitivity | 143 | 142 | 141 | 116 | 140 | 22 | 68 | 139 | 19 |

In this study, pressure changes due to artificial leakage were measured at four arbitrary places. At first, the monitoring location was determined by the two methods. Pressure changes due to leakage were measured to confirm the results of the monitoring location. Therefore, pressure contribution analysis and sensitivity analysis were performed using the measured pressure change. From the results of the analysis, the first rank of monitoring location was J-116, and the second, third, and fourth were J-140, J-22, and J-68, respectively. Therefore, the first combination of monitoring location is J-116, J-140, J-22, and J-68. The leakage was produced at J-55 and pressure was measured at J-116, J-140, J-22, and J-68. Five different combinations of monitoring locations were used to measure the pressure at four different places. Figure 5 shows Combination 1 of the monitoring locations in the pilot plant. The leakage point was fixed as J-55. Leakage amount was controlled by a valve at J-55 from 0.00021 m$^3$/s to 0.00047 m$^3$/s. Combination 1 is the group of the top ranked junctions such as J-116, J-140, J-22, and J-68. The line pressure upstream was maintained as 9 m (88,200 N/m$^2$). After fixing the leakage location at J-55, five combinations of four pressure gauges measured the pressure changes according to artificial leakage. The nearest junctions to the leakage location are J-68, J-27, and J-70. Therefore, combinations are equally combined to avoid the combination of nearest junctions. Table 11 shows the pressure change in different combinations due to leakage at J-55. It is observed that Combination 1 shows the highest number for both pressure contribution and pressure sensitivity.

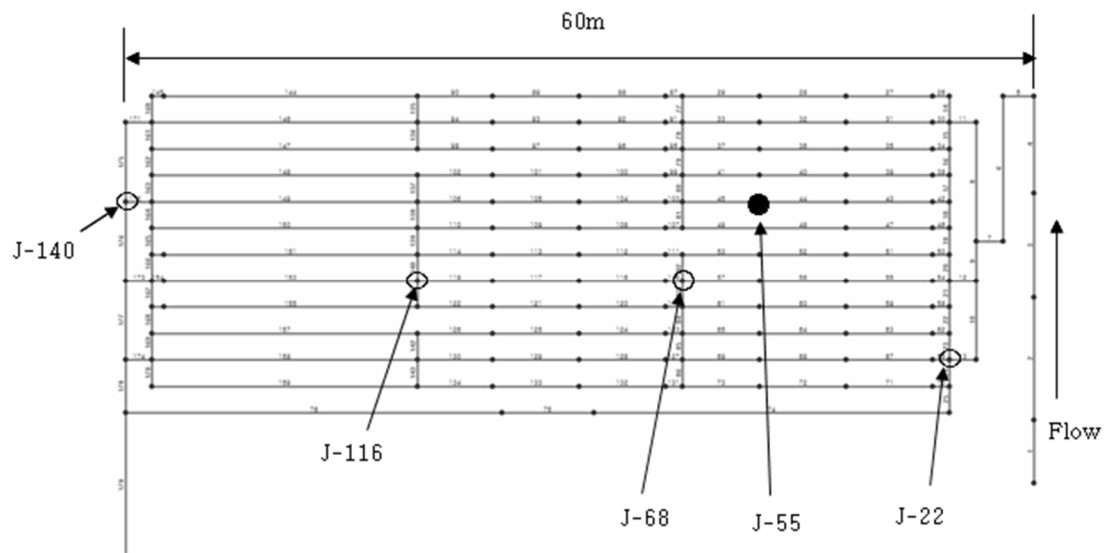

**Figure 5.** Plan view of pilot plant.

**Table 11.** Results of pressure contribution and sensitivity measurement.

| Combination Number | Sensor Location (Junction Number) | Rank for Pressure Sensitivity Analysis | Rank for Pressure Contribution Analysis |
|---|---|---|---|
| 1 | 22, 68, 116, 140 | 1 | 1 |
| 2 | 27, 70, 74, 113 | 3 | 2 |
| 3 | 70, 74, 114, 119 | 4 | 5 |
| 4 | 27, 70, 114, 119 | 2 | 3 |
| 5 | 68, 113, 134, 140 | 5 | 4 |

Therefore, combinations are equally combined to avoid the combination of nearest junctions. Table 11 shows the pressure change in different combinations due to leakage at J-55. It is observed that Combination 1 shows the highest number for both pressure contribution and pressure sensitivity.

*4.2. Test Bed*

A small block of H city was selected for the test bed as shown in Figure 6. In this area, 5000 people use this piping system, which contains 889 hydrants, 148 junctions, 162 pipes, and 1 distributing reservoir. The house is a one- or two-story private house that has a rooftop water tank. One distributing reservoir is only 15 m high in elevation and supplies 802 m³/day (0.00929 m³/s) of water. This pipe network consisted of a 15-year old Polyvinyl Chloride (PVC) or Polyethylene (PE) pipe with a pipe diameter of 0.05–0.2 m. The length of the longest pipe is 298 m. Pressure contribution analysis was conducted for the test bed. Twenty-five junctions out of 148 junctions were selected and analyzed by changing the demand at 25 junctions, as shown in Figure 6. The original demand at each 25th junction was changed by an additional demand of 0.0005 m³/s. From the results, it was found that the top ranked junction of pressure contribution is J-138. As shown in Table 12, J-139, J-148, J-110, and J-106 are the second, third, fourth, and fifth ranked junctions, respectively.

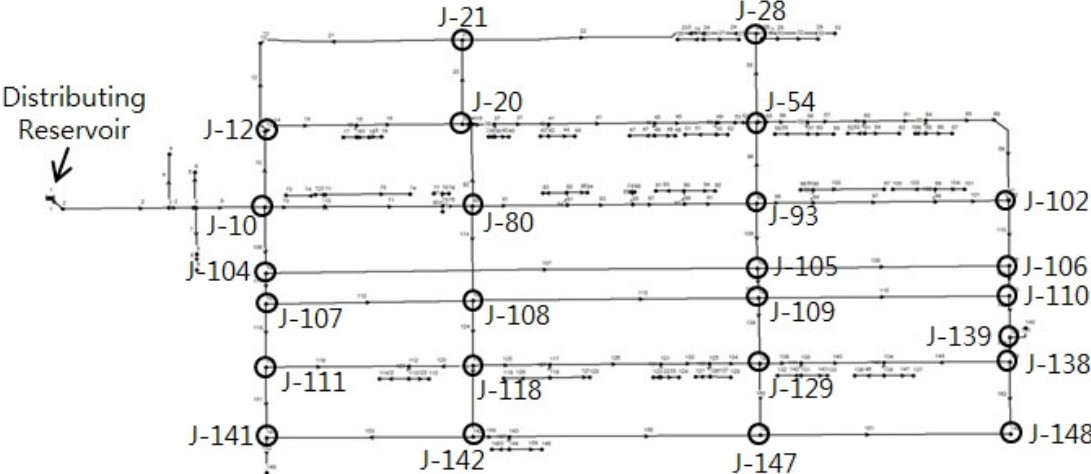

**Figure 6.** Small block pipe network of test bed.

**Table 12.** Results of pressure contribution analysis for test bed.

| Rank | 1 | 2 | 3 | 4 | 5 | 6 | 7 | 8 | 9 | 10 |
|---|---|---|---|---|---|---|---|---|---|---|
| Junction | 138 | 139 | 148 | 110 | 106 | 141 | 102 | 28 | 21 | 147 |

Pressure contribution analysis was conducted with the same conditions of analysis for the test bed. From the results, it was found that the top ranked junction of pressure contribution ratio is J-148. As shown in Table 13, J-138, J-139, J-110, and J-21 are the second, third, fourth, and fifth ranked junctions, respectively. From the results, it was found that the results of the sensitivity analysis are similar to the results of the pressure contribution analysis.

**Table 13.** Results of sensitivity analysis for test bed.

| Rank | 1 | 2 | 3 | 4 | 5 | 6 | 7 | 8 | 9 | 10 |
|---|---|---|---|---|---|---|---|---|---|---|
| Junction | 148 | 138 | 139 | 110 | 21 | 106 | 147 | 102 | 20 | 28 |

## 5. Conclusion

In the present study, two determination methods of monitoring locations in a water distribution system were developed and applied to a pilot plant and a real test bed. The proposed methods were the pressure contribution analysis and sensitivity analysis, which were applied to the pilot plant and compared with the experiments for verification. From the results, it was found that the top ranked monitoring location was concentrated to the downstream of the pilot plant piping system. Therefore, the fourth, fifth, sixth, and seventh ranked monitoring locations, namely J-116, J-140, J-22, and J-68, were chosen for the monitoring location. To verify the methods, leakage at J-55 was artificially produced at the pilot plant. The pressure change was measured at five different combination groups of sensor locations. From the results, it was found that the top ranked group of sensor locations, J-116, J-140, J-22, and J-68, had the highest pressure contribution and sensitivity. The pressure contribution analysis and sensitivity analysis were applied to a small sample piping system. To verify the two methods, unsteady analysis was conducted. The results of the proposed method were similar to the results of unsteady analysis. Finally, the two methods were applied to a real distribution system of a small city as a test bed. It was verified that the newly developed methods of monitoring locations were useful and effective. These two methods can be applied to determine the pressure monitoring location and to determine where a hydraulic device can be installed in a distribution system for the operation and management of the water supply system. For the future study, two methods for monitoring location in a water distribution system will be tested by installing pressure sensors in the real water distribution system.

**Author Contributions:** Conceptualization, H.J.K. and S.J.; Methodology, H.J.K.; Investigation, H.J.K.; Writing—Original Draft Preparation, H.J.K. and S.J.

**Funding:** This research received no external funding.

**Conflicts of Interest:** The authors declare no conflict of interest.

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
