# Peer review of "The Optimum Monitoring Location of Pressure in Water Distribution System"

_water, doi:10.3390/w11020307_

Round 1
Reviewer 1 Report
Determining pressure monitoring location is important technique for optimal operation and management of water distribution system. The reviewer believe that the presented results show the possibility of future application. However, the reviewer has several questions that the authors should address in the revision for publication.
This study proposed two methodologies which determine the optimum pressure monitoring locations. It would be better if the authors can suggest the better one since the engineers need to know which method can be selected according to network characteristics such as size and shape of network.
This study focuses on pressure measurements in water distribution system. Therefore, I think the title could be changed accordingly.
Draft explained very well about reasons why we determine the number and location of gauges. But it doesn’t explain why we should install the pressure gauges.
Table 7 doesn’t have unit.
In chapter 2, it would be better to specify how much the water demand was changed.
Author Response
Authors really appreciate for reviewer’s comments. And most of suggestions have been accepted and revised.
Determining pressure monitoring location is important technique for optimal operation and management of water distribution system. The reviewer believe that the presented results show the possibility of future application. However, the reviewer has several questions that the authors should address in the revision for publication.
This study proposed two methodologies which determine the optimum pressure monitoring locations. It would be better if the authors can suggest the better one since the engineers need to know which method can be selected according to network characteristics such as size and shape of network.
- Authors think that both two methods can be recommended at the same time no matter what network looks like. Results of simulations and computations show also similarity. However, authors can cautiously recommend sensitivity analysis because it is little bit easier to use.
This study focuses on pressure measurements in water distribution system. Therefore, I think the title could be changed accordingly.
- Yes, title shows now “pressure”.
Draft explained very well about reasons why we determine the number and location of gauges. But it doesn’t explain why we should install the pressure gauges.
- Water pressure in distribution system is very important matter since we cautiously monitor the pressure at the end of pipeline. Enough pressure has to be exist at the end of pipeline. Therefore, monitoring the pressure in pipeline is very important in matter of management and maintenance of water distribution system. This is inserted in main text (page 1-2)
Table 7 doesn’t have unit.
- Yes, unit is inserted in table.
In chapter 2, it would be better to specify how much the water demand was changed.
- Amount of change of demand is 0.02m3/s. It was inserted in manuscript (page 6).

Reviewer 2 Report
This study proposed two different methods for the determination of monitoring location for pressure gauges. It means a lot since two methods were applied to real network and pilot plant.
The topic of this study is within the scope of the Journal. However, some explanations and errors in equations make it difficult to understand the contents of this study. Therefore, I recommend that the authors accept the below comments to increase the completeness of this draft, I hope my suggestion can help.
Major
1. Authors need to define the amount of change of demand in chapter 2. And is there any rule of thumb for that? Please explain it.
2. Recent references have to be discussed for related this topic. For example,
- Symeon E. Christodoulou, Anastasis Gagatsis, Savvas Xanthos, Sofia Kranioti, Agathoklis Agathokleous, Michalis Fragiadakis (2013). “Entropy-based sensor placement optimization for waterloss detection in water distribution networks.” Water Resources Management, Published online 05/Sep. 2013.
3. For the conclusion, is two methods have same results? Can we determine which one is better than the other?
4. Table 7 needs unit.
5. Figure 5, 6 are replace the high-resolution figure.
6. The authors add the contents of the [Autor Contributions], [Funding], and [Acknowledgment] as following example:
- [Autor Contributions]
Xxx xxx wrote the draft of the manuscript and provided this model. Xxx xxx and Yyy yyy carried out the survey of previous studies. Xxx xxxx and Yyy yyy conceived the original idea of the proposed method.
- [Funding], or [Acknowledgment]
The study was supported by OOO OOO Institute as “Projects for XXX XXX (2010123123: Project number)”.

Author Response
Authors really appreciate for reviewer’s comments. And most of suggestions have been accepted and revised.
This study proposed two different methods for the determination of monitoring location for pressure gauges. It means a lot since two methods were applied to real network and pilot plant.
The topic of this study is within the scope of the Journal. However, some explanations and errors in equations make it difficult to understand the contents of this study. Therefore, I recommend that the authors accept the below comments to increase the completeness of this draft, I hope my suggestion can help.
Major
1. Authors need to define the amount of change of demand in chapter 2. And is there any rule of thumb for that? Please explain it.
- Amount of change of demand is 0.02m3/s. It was inserted in manuscript (page 6).
2. Recent references have to be discussed for related this topic. For example,
- Symeon E. Christodoulou, Anastasis Gagatsis, Savvas Xanthos, Sofia Kranioti, Agathoklis Agathokleous, Michalis Fragiadakis (2013). “Entropy-based sensor placement optimization for waterloss detection in water distribution networks.” Water Resources Management, Published online 05/Sep. 2013.
3. For the conclusion, is two methods have same results? Can we determine which one is better than the other?
- Authors think that both two methods can be recommended at the same time no matter what network looks like. Results of simulations and computations shows also similarity. However, authors can cautiously recommend sensitivity analysis because it is little bit easier to use.
4. Table 7 needs unit.
- Yes, unit is inserted in table.
5. Figure 5, 6 are replace the high-resolution figure.
- Yes, Figures are replaced.
6. The authors add the contents of the [Autor Contributions], [Funding], and [Acknowledgment] as following example:
- [Autor Contributions]
Xxx xxx wrote the draft of the manuscript and provided this model. Xxx xxx and Yyy yyy carried out the survey of previous studies. Xxx xxxx and Yyy yyy conceived the original idea of the proposed method.
- [Funding], or [Acknowledgment]
The study was supported by OOO OOO Institute as “Projects for XXX XXX (2010123123: Project number)”.
- Authors really appreciate your kind recommend.

Reviewer 3 Report
Authors proposed two methods for optimal location of pressure monitoring to assess leakage in the water distribution networks. I have following observations to bring to the notice of authors.
Comment: Abstract: please present qualitative results of proposed methods applied to case studies.
Comment 1: Authors should state why it is impossible to locate pressure gauge at all the nodes.
Comment 2: How to extend sensitivity analysis if two or more pipes undergo change in Chw?
Comment 3 Change in Chw affects the pressure at several nodes. And it is ranked based on proposed sensitivity index. What is effect of pipe size on these changes? No doubt, the rank could be different if pipe size and its length are taken into the account.
Comment 4: In Fig. 2 some text is missing. Please correct it.
Comment 5: What about authors’ recommendation on the method of optical location of monitoring pressure? Authors should state which is one better? Though the study provides close results, authors should highlight which method will be more appropriate.
Comment 6:'Conclusions' comprises general issues about the problem as explained before within the text. I expect the author to focus on the findings and results of the study.
Comment 7: In the conclusion, authors are specifying optimal location in terms of Junction ID. Is this common for all the networks?
Comment 8: Conclusions: Authors did not explain why downstream nodes are more sensitive in pressure changes.
Comment 9: Conclusions should explain finding clearly from the study and provide framework for future research.
Author Response
Authors really appreciate for reviewer’s comments. And most of suggestions have been accepted and revised.
Authors proposed two methods for optimal location of pressure monitoring to assess leakage in the water distribution networks. I have following observations to bring to the notice of authors.
Comment: Abstract: please present qualitative results of proposed methods applied to case studies.
- Yes, some of conclusions is now inserted in abstract as you pointed out.
Comment 1: Authors should state why it is impossible to locate pressure gauge at all the nodes.
- Pressure sensors are kind of expensive and very difficult to manage. Therefore, we should carefully think about where we are going to install it. That’s why authors state it is very important.
Comment 2: How to extend sensitivity analysis if two or more pipes undergo change in Chw?
- Of course, if we change the Chw of two or more pipes, rank will be change. However, we will be confused which pipe affects the results of pressure change. That’s why authors simulated one by one.
Comment 3 Change in Chw affects the pressure at several nodes. And it is ranked based on proposed sensitivity index. What is effect of pipe size on these changes? No doubt, the rank could be different if pipe size and its length are taken into the account.
- Yes, it will be changed. However, rank results of this study is applied for the sample network Fig. 1. Because simulations are based on only Fig. 1 in this study.
Comment 4: In Fig. 2 some text is missing. Please correct it.
- Yes, it was corrected.
Comment 5: What about authors’ recommendation on the method of optical location of monitoring pressure? Authors should state which is one better? Though the study provides close results, authors should highlight which method will be more appropriate.
- Authors think that both two methods can be recommended at the same time no matter what network looks like. Results of simulations and computations shows also similarity. However, authors can cautiously recommend sensitivity analysis because it is little bit easier to use.
Comment 6:'Conclusions' comprises general issues about the problem as explained before within the text. I expect the author to focus on the findings and results of the study.
Comment 7: In the conclusion, authors are specifying optimal location in terms of Junction ID. Is this common for all the networks?
- It is not common for all the network. But authors want to explain detail results of experiment on pilot plant.
Comment 8: Conclusions: Authors did not explain why downstream nodes are more sensitive in pressure changes.
- Valves are usually located at the end of pipeline. It is very sensitive location for pressure change with valve activity.
Comment 9: Conclusions should explain finding clearly from the study and provide framework for future research.
- For the future study, two methods for monitoring location in water distribution system should be tested by installing pressure sensors in the really size of water distribution system. Furthermore, this study also can be applied to the leakage detection method. This is inserted in Conclusion.
Round 2
Reviewer 2 Report
This study proposed two different methods for the determination of monitoring location for pressure gauges. It means a lot since two methods were applied to real network and pilot plant.
The topic of this study is within the scope of the Journal and well written. From the 1st revision, the authors accepted and modified the comments of reviewers appropriately. There, I recommend that this paper can be accepted for publication.
Reviewer 3 Report
Nil